

# Longitudinal survey data for diversifying temporal dynamics in flood risk modelling

Elena Mondino[1,2], Anna Scolobig[3], Marco Borga[4], Giuliano Di Baldassarre[1,2]

[1]Department of Earth Sciences, Uppsala University, Uppsala 752 36, Sweden
5  [2]Centre of Natural Hazards and Disaster Science (CNDS), Uppsala 752 36, Sweden
[3]Environmental Governance and Territorial Development Institute, University of Geneva, Geneva 1205, Switzerland
[4]Department of Land, Environment, Agriculture and Forestry, University of Padua, Padua 351 22, Italy

*Correspondence to*: Elena Mondino (elena.mondino@geo.uu.se)





**Abstract.** Numerous scholars have unravelled the complexities and underlying uncertainties of coupled human and water systems in various fields and disciplines. These complexities, however, are not always reflected in the way in which the

35    dynamics of human-water systems are modelled. One reason is the lack of social data times series, which may be provided by longitudinal surveys. Here, we show the value of collecting longitudinal survey data to enrich sociohydrological modelling of flood risk. To illustrate, we compare and contrast two different approaches (repeated cross-sectional and panel) for collecting longitudinal data, and explore changes in flood risk awareness and preparedness in a municipality hit by a flash flood in 2018. We found that risk awareness has not changed significantly in the timeframe under study (one year).

40    Perceived preparedness also did not change, but we observed differences related to damage severity. More precisely, preparedness increased only among those respondents who suffered low damages during the flood event. We also found gender differences across both approaches for most of the variables explored. Lastly, we argue that results that are consistent across the two approaches constitute robust data that can be used for the parametrisation of sociohydrological models. We posit that there is a need to improve socio-demographic heterogeneity in modelling human-water systems in order to better

45    support risk management.





# 1 Introduction

Over the past decades, numerous scholars have engaged with the study of coupled human and water systems in various research fields and disciplines, including e.g. environmental history, sociology and philosophy, as well as ecological economics, philosophy of science, social-ecological systems and sociohydrology (e.g. Aldrete, 2007; Di Baldassarre et al., 2019; Folke et al., 2005; Hoffmann et al., 2020; Kallis & Norgaard, 2010; Liu et al., 2007; Ostrom, 2009; Schlüter et al., 2012; Sivapalan et al., 2012). Many of these studies adopt a system thinking approach to embrace the complexities and underlying uncertainties of natural systems and the way in which human systems affect and are affected by them (Checkland, 2000; Checkland & Poulter, 2006).

Falling under the bigger umbrella of socio-ecological systems (SES, Redman et al., 2004), sociohydrological systems are also the result of a system thinking approach aimed at unravelling the complex interactions between people and water, specifically. Recently, sociohydrological models have emerged as useful tools to explain risks generated by feedback mechanisms between social and hydrological systems, and thus aim at supporting hydrological risk policy processes (Blair & Buytaert, 2016). Many scholars have developed sociohydrological models of flood risk, using either system dynamics of lumped society (e.g. Barendrecht et al., 2019; G. Di Baldassarre et al., 2013; Liu et al., 2017; Viglione et al., 2014) or agent-based approaches (e.g. Haer et al., 2019; Michaelis et al., 2020). These models include social parameters describing for example attitudes and behaviours towards risk. Yet, the inclusion of complex social dynamics may increase structural uncertainty and open up questions about the reliability of modelling (De Marchi, 2020; Saltelli & Funtowicz, 2015).

The IPCC (Metz et al., 2007) argues that structural uncertainty is reduced when a) convergent results are obtained using different methods, and b) results rely on empirical data rather than calculations. Hence, to make sure that sociohydrological models serve their purpose, they must employ different types of data. Moreover, these data should be robust. While time series concerning physical aspects, such as precipitation, runoff, and flood water levels are often available, data availability is limited when it comes to time series concerning social parameters, such as changes in flood risk awareness and preparedness over time (Barendrecht et al., 2019; Mondino et al., 2020a). Sociohydrological models of flood risk use changes in awareness as primary mechanisms explaining the emergence of unintended consequences, such as the safe-development paradox (e.g. Barendrecht et al., 2019; G. Di Baldassarre et al., 2013; Liu et al., 2017; Viglione et al., 2014). Thus, the empirical social data with the greatest contribution potential are longitudinal survey data (Bubeck et al., 2020; Hudson et al., 2020; Siegrist, 2013, 2014; Sivapalan, 2015). Indeed, the availability and use of longitudinal psychological and behavioural data is key to reduce structural uncertainty within sociohydrological modelling (Di Baldassarre et al., 2016).

Specifically, longitudinal survey data on risk awareness and preparedness are required to explore how a sociohydrological system evolves over time in the presence or absence of hydrological extremes (e.g. floods or droughts), as such data represent potential drivers of behavioural change. Knowing if and how people's awareness of risk changes over time and



how people may or may not be prepared for a disastrous event, e.g. by adopting private protection measures or by supporting the implementation of structural or non-structural risk reduction measures, is fundamental to better understand human impacts on the water system. Moreover, and most importantly, a better understanding of preparedness dynamics can save

lives. In fact, this knowledge may, among others, uncover potentially heterogeneous adaptation trajectories. When robust and reliable, this knowledge may also contribute to identify social data proxies that may be helpful to evaluate the long-term effectiveness of risk awareness and communication campaigns, hence helping to overcome the over-simplified representation of community as homogeneous community, where every individual acts, reacts, and thinks in the same way.

Excluding Barendrecht et al. (2019), no socio-hydrological model exploring human-flood interactions has employed

empirical longitudinal survey data. Barendrecht et al. (2019) used empirical data to estimate the parameters of a sociohydrological flood risk model by means of Bayesian inference. The longitudinal survey data was collected in Dresden, Germany after the flood events of 2002, 2006, and 2013 (more information in Kreibich et al., 2005; Kreibich & Thieken, 2009; Thieken et al., 2016). They argue that, while both sociohydrological models and empirical studies have their own limitations, the combination of the two may help bring out their pros while mitigating their cons. In their study, they

conducted a sensitivity analysis to unravel the influence of a number of variables on a simple sociohydrological model. They found that when risk awareness data are absent, most of the parameters estimations are biased and, even more worrying, the modelled dynamics between the water and the human system over time are wrong. Concerning preparedness, their analysis shows that a lack of preparedness data does not have an impact as serious as the lack of awareness data, because it does not influence the estimation of other parameters. They also add that in absence of such data the dynamics of preparedness itself

may be biased. On one hand, this may be due to how preparedness was defined in their model (a ratio of protection measures taken by a household versus the total amount of protection measures available). On the other, this undoubtedly shows the paramount importance of collecting longitudinal data on risk awareness and preparedness, especially within the field of flood risk, to avoid modelling errors and biases that would be otherwise difficult to identify. An effort in this direction was recently made by Ridolfi et al. (2020), who explored the influence of collective flood memory on flood losses, even though

the empirical data consisted in proxy information about flood memory, i.e. archaeological data about the average vertical distance of human settlements from the river, rather than longitudinal surveys.

Moreover, despite the need for empirical longitudinal data per se (Siegrist, 2013), there is a need for reliable and robust data on attitude and behavioural dynamics, such as how risk awareness and preparedness change over time. The general assumption in the literature is that, after the occurrence of a flood and in absence of consequent flood events, risk awareness

decreases over time. This assumption finds its roots in the fact that sociohydrological models of flood risk often use memory as a proxy for risk awareness (Di Baldassarre et al., 2013; Viglione et al., 2014). Besides the fading of memory itself, the use of memory is inevitably connected to cognitive processes (e.g. the availability heuristic, Tversky & Kahneman, 1973) asserting that people tend to judge the probability and consequences of an event based on the ease with which it comes to





mind. Following this reasoning, after the occurrence of a flood and in absence of consequent flood events, risk awareness
should decrease. Recently, Bubeck et al. (2020) and Mondino et al. (2020a) provided evidence on the decreasing rate of risk
awareness and preparedness in the aftermath of a flood event with empirical survey data. However, these studies are just one
step forward, and more evidence is needed in terms of attitude and behavioural dynamics. Indeed, the more robust is such
data, the easier and more precise is their parametrisation in sociohydrological models. As mentioned above, a way to reduce
structural uncertainty and obtain reliable data is to stress-test the variables of interest using different methodologies. In this
study, we present and compare two methods to collect longitudinal data:

    i.    *repeated cross-sectional* approach, consisting of conducting cross-sectional surveys two (or more) times over the
years in the same area;

    ii.    *panel* approach, consisting in surveying exactly the same individuals two (or more) times over the years.

We argue that consistent results about the change (or lack thereof) in risk awareness and preparedness will not only provide
robust data and additional evidence to be employed in sociohydrology, but also inform disaster risk communication
strategies and policies. We illustrate the two approaches with a case study of a village in North-eastern Italy that was hit by a
flash flood in 2018. Two survey rounds were conducted. The first one in February 2019, 6 months after the flood event, and
the second one in February 2020, 18 months after the flood event.

## 2 Background

Depending on the information we are interested in, survey data can be collected in two different ways: via a cross-sectional
approach or via a longitudinal approach. A *cross-sectional design* can be defined as a picture, a snapshot, which provides us
with information about a certain variable at a specific point in time and space, and thus consists of just one sample surveyed
at one point in time. Cross-sectional studies are optimal when the researcher is not interested in detecting a change.
However, the snapshot approach is not always informative of a process that is ongoing and constantly changing. Things that
happened in the past shape current behaviours, and they must be taken into account. To this end, it would be optimal to have
a research approach that provides us with a 'video' of the process, but since this is obviously not feasible, the best option is
to take multiple pictures over time and detect potential differences. This is what a *longitudinal design* does (Payne & Payne,
2011). Longitudinal studies consist in surveying either two or more different samples collected at two or more points in time
(repeated cross-sectional) or the same sample at two or more points in time (panel). Therefore, a longitudinal design helps
not only spot changes in comparison to previously recorded perceptions and behaviours, but also recognise any correlation
between variables, as well as to avoid misleading conclusions (Siegrist, 2013). In fact, potentially misleading results from
cross-sectional studies could end up in wrong policy recommendations. In light of these characteristics, the need for
longitudinal data has been highlighted by a number of scholars in the natural hazards field (Babcicky & Seebauer, 2017; Di



Baldassarre et al., 2018; Fielding, 2012; Lindell & Perry, 2000; Spence et al., 2011; Terpstra, 2011; van Duinen et al., 2015).
However, despite a large majority of scholars within sociohydrology acknowledge this lack of longitudinal data, the majority of empirical studies within the flood risk domain adopt a cross-sectional approach (a review in Kellens et al., 2013), and empirical studies that adopt a longitudinal approach are still rather scarce (Barendrecht et al., 2019; Hudson et al., 2020).

The following sections will go through the two main types of longitudinal studies, repeated cross-sectional and panel studies. The difference between the two lays mostly in the sampling procedure and the individuals who are sampled, but this
significantly influences the type of statistical analysis that can be conducted on the respective data and the pros and cons of each approach.

## 2.1 Repeated cross-sectional studies

A repeated cross-sectional study (RCS) consists in repeating the same survey over time without necessarily involving the same respondents. This is similar to a cohort study, which consists in sampling individuals with a shared characteristic,
which makes them part of a "cohort", e.g. being born in the same year, living in the same town, having experienced a certain event, and so forth. Therefore, the two (or more) samples taken over time contain different individuals at different points in time. There are studies adopting a mixed sampling, i.e. the two samples may contain some of the same individuals, but not all (Kienzler et al., 2015). However, if not properly accounted for, this may lead to issues in the statistical analysis. If we sample inhabitants of a town in 2019 and then we sample other inhabitants of the same town in 2020, then we are adopting a
repeated cross-sectional approach. This approach allows the sample to keep its size over time, as there is no need to recruit the same individuals in the consecutive survey rounds. This makes the approach rather resource friendly. As a consequence, by adopting this approach, the possibility of following the evolution of an individual over time is lost, and comparisons can only be made between clusters of individuals. Thus, changes can only be analysed at the societal level. An example of this approach is presented in Mondino et al. (2020a), who studied changes in risk awareness and preparedness in two
municipalities in the North-eastern Italian Alps. Salvati et al. (2014), conducted a similar study in Italy, but at the national level, where they explored risk perception for a number of natural and technological hazards (among which floods). They conducted two rounds of surveys in 2012 and 2013 and found a slight decrease in the percentage of respondents with a high flood risk perception, but no statistical significance was reported. RCS studies are also often conducted at the international level, to assess differences among countries regarding certain indicators. An example is the Labour Force Survey (Eurostat),
which is conducted in various countries and aims at collecting data on the labour market, such as e.g. the unemployment rate.

## 2.2 Panel studies

A panel study investigates a sample composed of the same individuals over time, e.g. 100 people who agree to take part in a certain research and be assayed multiple times in a 15-year time span. If we want to adopt a longitudinal approach, following



the previous example we must interview in 2020 the same individuals whom we interviewed in 2019, provided that they
accept to partake in the survey again. This approach is optimal when we are interested in following the evolution of an
individual over time. It also allows for more in depth statistical analyses, as it is possible to introduce the random effect of
the individual into regression models. On the other hand, maintaining a longitudinal panel is resource-intense, as the
panellists' interest declines over time, especially when it comes to flood risk. This is because the initial sample is smaller
than in other cases, given that floods are usually locally confined (Hudson et al., 2020). The loss of panellists over time is
known as *attrition rate*. If the panellists drop out non-randomly, i.e. they share certain characteristics, then we incur into
*attrition bias* (however, there is little evidence of that in the flood risk domain, Hudson et al., 2020). When the attrition rate
is considerably high, we incur into retention bias, i.e. the number of observations is too small that any statistical analysis
loses significance.

Hudson et al. (2020) discussed the challenges of longitudinal surveys in the flood risk domain. In their review, they collected
all studies that adopted a panel approach within the flood risk domain, and then explored the potential for attrition rate, bias,
and retention bias. They then explored the same in a panel study conducted in Germany after the flood event in 2013. They
found little evidence for attrition bias. As for the attrition rate, this varies greatly among studies. The average in their
literature review was 38%, while in their panel study in Germany it had an average per wave of 60%. In general, they found
that studies using pre-existing panels (e.g. the Swiss Household Panel Study, FORS) have a much lower attrition rate
compared to panels specifically developed to investigate a flood-affected population.

In their review, Hudson et al. (2020) found only seven empirical studies that followed a panel approach within the flood risk
domain up to 2018, in addition to their own (Calvo et al., 2015; Fay-Ramirez et al., 2015; Fothergill, 2003; Ginexi et al.,
2000; Kaniasty & Norris, 2008; Lin et al., 2017; Osberghaus, 2017). Since 2018, we found four other studies that adopted
such approach in the flood risk domain. Bodoque et al. (2019) investigated changes in risk perception after introducing a risk
communication strategy 8 months after the first survey round in a municipality hit by a flash flood in Spain. After
conducting a second survey round one month after the implementation of the strategy, they found that those respondents who
were exposed to risk communication activities had a slightly higher risk awareness, but only when thinking about the town
as a whole. The effect of risk communication strategies in promoting mitigation behaviour was also explored by Osberghaus
and Hinrichs (2020), but they did not investigate changes in risk awareness itself. Bubeck et al. (2020), using the same
dataset employed in Hudson et al. (2020), investigated dynamics of human behaviour in response to flooding, following
individuals over three survey rounds 9, 18, and 45 months after the 2013 flood in Germany. They only detected a decrease in
risk perception in terms of perceived probability between the second and third wave. Seebauer and Babcicky (2020) explored
causal relationships within the Protection Motivation Theory in Austria, but they did not detect any statistically significant
difference in risk perception between the two survey rounds (15 months apart).





Following the effort of Hudson et al. (2020), we here explore the differences in results yielded by panel and repeated cross-sectional data, collected with the same survey in a single study area. No study so far has investigated this difference on the basis of empirical data within the flood risk domain. The next section will describe in detail the sampling procedures.

## 3 Methods

### 3.1 Study area

The municipality of Negrar, located in the Veneto Prealps north of Verona, served as a case study. The administrative area ranges from 70 to 860m a.s.l. Three main streams (locally known as *progni*) flow through the municipality and merge with the Adige river downstream. Negrar is further divided into smaller urban conglomerates, mainly located in the floodplains in the southern part of the municipality. Its population steadily increased in the last years and reached 16.850 units as of 2020.

On September 1st, 2018, one of the small urban conglomerates (Arbizzano-Santa Maria) was hit by a flash flood. Heavy
rainfall accumulated more than 180mm in less than three hours and the *progno di Novare* overflowed, flooding the nearby buildings with a peak discharge that reached 20m$^3$/s. Because of its intensity, this flash flood – characterised by a 100-year return period – caused severe economic damages (~10 million Euros) and affected more than 3000 people (although no casualties were reported). Data collection was thus focused on Arbizzano-Santa Maria (4000 inhabitants). Further information on the hydrology of the area are presented in Weyrich et al. (2020) and Mondino et al. (2020c), while
comprehensive hydrological data and their collection are presented in Amponsah et al. (2018) and Borga et al. (2019).

### 3.2 Data collection

Data collection was based on a questionnaire survey carried out twice face-to-face in February 2019 and February 2020. Between the two surveys, the local administration organised a number of events to inform the residents on various aspects of the 2018 flood. In one event, a local meteorological association explained the potential causes that led to the flooding. In
another event, the municipality hosted one of the authors of this paper to present the results following the first round of interviews (presented in Weyrich et al., 2020 and Mondino et al., 2020c). In addition to increasing the maintenance of the smaller streams running through the town (which overflowed causing the 2018 flooding), the municipality also undertook the construction of a flood diversion channel to redirect the water coming from the streams towards a larger stream that eventually flows into the Adige river. This way, the amount of water in the smaller streams is reduced and so is the
probability of overflow during heavy rain periods.

In the first survey round, we favoured a stratified sample of residents based on quotas (Stockemer, 2019) over a random sample, which might have resulted in the exclusion or underrepresentation of those residents living in the most risky areas and the households most affected by the 2018 flood. The sample was stratified according to age and gender (based on Italian





National Census data). Each interviewer (six in total) was provided with a grid containing the target distribution of
interviewees, to respect the statistical distribution of these variables in the local population. The demographic data were
provided by the Civil Registry of Negrar, and the administration also provided a list of residents that were affected by the
flash flood who agreed to be interviewed. The interviewers were then instructed to first contact the people on the list via
phone calls to set appointments for the interviews, a fundamental preliminary step to establish trust in the local community.
In turn, the residents who were interviewed first helped the interviewers establish trust with neighbours in order to facilitate
the interviewing process in households affected by the 2018 flood and avoid unnecessary nuisance. In addition, the
interviewers received a map of the study area and were instructed to contact each household in each and every street, to
maximise randomisation and, at the same time, fill in the quotas that were required for age and gender. The unit of analysis
was the individual, and interviewees were instructed to interview only one person per household. The restriction to one
person per household was due to the presence of questions relating to the adoption of protection measures within the
household, which are not discussed in the present paper but are extensively presented in Weyrich et al. (2020). Thus,
interviewing more than one person per household would have led to duplicates for the questions about the adoption of
protection measures.

Data in the first survey round were collected between February 18th and March 1st, 2019, approximately six months after the
flood event. Local authorities approved the survey. Participants received no incentive to complete the survey, which took
them on average 30 minutes to complete. At the end of the interview, respondents were asked whether they agreed on being
contacted again a year later to fill in the same questionnaire. If they agreed, they were asked to provide a contact (phone
number or email address), so that they could be contacted again.

Data in the second survey round were collected between February 17th and March 1st, 2020. Here, we drew two samples. The
first sample was drawn in the same way as the original sample, e.g. stratified sample based on quotas and representative of
the local population in terms of age and gender. Interviewers were instructed not to include in this sample those respondents
who completed the survey in the first round. This sample, together with the original, constitutes the RCS dataset.

The second sample consisted exclusively of individuals who participated in the first survey round and who accepted to
participate again. This second sample, together with the original one, constitutes the panel dataset. Summary statistics for all
three samples are presented in Table 1.

**3.3 Variables assayed**

The variables selected to explore flood risk awareness fall under five categories: general feeling of safety, flood threat, flood
damage, knowledge, and trust. The variables selected to explore preparedness investigated the perceived individual
preparedness, the adoption of private protection measures, and the stipulation of a flood insurance.



The variables, the related questions, and available answers are listed in Table 2, and the complete survey questionnaire can
be found in the supplementary materials.

Regarding the damage suffered during the 2018 flood, to facilitate the analysis respondents were divided into three groups:
those who did not experience any damage (i.e. 1 on the 1, min to 5, max scale), those who experienced low damage (i.e. 2-3
on the scale), and those who experienced high damage (i.e. 4-5 on the scale).

The statistical analysis conducted on the RCS dataset consisted of single regressions using cumulative link models (clm).
Because the panel approach consists in following the individual respondent over time, giving us an insight into how
differently or similarly the same person replied to the same questions posed to them, the statistical analysis conducted on the
panel dataset consists of single regressions using cumulative link mixed models (clmm), which allow for the introduction of
random effects (in this case, the individual respondent). Both analyses were conducted with the software for statistical
computing R (version 3.5.2), using the package "ordinal" (Christensen, 2019). We adopted a 90% confidence interval.

Before presenting the results of the two analyses concerning changes in risk awareness (4.1) and changes in preparedness
(4.2), the following section will discuss the potential for attrition bias and retention bias in the panel dataset.

### 3.4 Attrition bias and retention bias

In the first survey round, 86% of the respondents (N = 125) agreed to be contacted again, but only 58% of them (N = 84)
actually repeated the survey in the second round, leading to a 42% attrition rate. This is in line with the average 40% attrition
rate reported in other studies using specifically developed surveys (Hudson et al., 2020). While this percentage is not
particularly high per se, when considering the relatively small size of the first sample it may create some issues regarding the
statistical significance of the analysis conducted, i.e. there is potential for retention bias and this must be taken into account
when interpreting the results.

In addition, we ran an ordinal logistic regression to assess the probability of a respondent moving from survey round 1 to
round 2 depending on the main variables connected to our research question (changes in flood risk awareness and
preparedness). This allows for exposing any potential attrition bias due to data missing at random (MAR, van Buuren, 2018),
i.e. when missing data (respondents who drop-out) are connected to observed factors. This is just one of the ways in which to
test for attrition bias. Other methods are used in the econometric literature (see e.g. Alderman et al., 2001; Little & Rubin,
2019).

Table 3 shows that only few of the variables that are important for our research question affect the probability of a
respondent to move to the next survey round. The variables *age* and *age squared* show a non-linear relationship, meaning
that as the age increases participation in the next round tends to increase, but at a decreasing rate. The same result was also





reported by Hudson et al. (2020). Concerning the *perceived threat posed to the town by floods*, participants who replied on the higher end of the scale in the first round are more likely to participate in the second. *Individual preparedness*, on the other hand, has the opposite effect: respondents who reported low levels of individual preparedness in the first round are more likely to participate again in the second.

In order to correct for the potential bias due to attrition, we conducted an Inversed Probability of Attrition Weighing (IPAW, Hernán & Robins, 2020). This procedure assigns weights to respondents depending on their probability of moving to the next survey round on the basis of the main variables of interest (those showed in Table 3). Respondents who share similar characteristics with those who dropped out after the first survey round are assigned a heavier weight, thus compensating for the loss of respondents. The weights are assigned only to respondents in round 2, as respondents in round 1 all have the same weight.

## 4 Results

A comparison of the panel analysis with and without IPAWs showed that the differences in results were negligible. Hence, the following section reports the results following the analysis without IPAWs, to reduce post hoc data manipulation to the minimum. The analysis conducted using IPAWs is reported in the supplementary material (Table S1).

Figure 1 shows the different effect of time on the variables of interest, as resulted from the clms and clmms.

### 4.1 Risk Awareness

The RCS approach shows that the general feeling of safety about living in the area decreased, but only for women (Odd Ratio = .61, Confidence Interval$_{90\%}$ = .38–.99). The panel approach shows no change over time. Respondents feel safe living in the area, with 72% of them replying 4 or above on the 1 min 5 max scale in round 1, 67% in round 2 – RCS, and 76% in round 2 – Panel.

#### 4.1.1 Perceived threat posed by floods

The perceived threat posed by floods to oneself, one's home, and the town as a whole, show inconsistent changes over time across the two approaches.

When it comes to the perceived threat posed by floods to oneself, both approaches show no change over time. However, when breaking down the sample, the RCS shows that it actually significantly increased for those respondents who suffered high damage during the 2018 flood (OR = 5.23, CI$_{90\%}$ = 1.99–13.9), but it decreased for those who did not suffer any damage (OR = .46; CI$_{90\%}$ = .27–.77). This difference was not found in the panel study. As for the perceived threat to the



home, in round 2 – RCS less women are concerned compared to round 1 (OR = .61, CI$_{90\%}$ = .38–.99), while the panel approach shows no change over time. The threat to the town as a whole did not seem to change over the entire sample in both approaches, but when breaking down the sample, the panel shows that it actually decreased for women (OR = .50, CI$_{90\%}$ = .27–.93). The panel also shows that women were more concerned than men in round 1 (OR = 2.61, CI$_{90\%}$ = 1.57–4.38), but this difference is lost in round 2 due to the decrease in perceived threat for women. Respondents with a higher income are

also less concerned for themselves in both years (RCS: OR = .69, CI$_{90\%}$ = .55–.86; Panel: OR = CI$_{90\%}$ = .62, .45–.84). Age was not found to affect any of the threat variables neither in the RCS nor in the panel approach. Education was found to play a role only in the RCS dataset, with more educated respondents showing a lower perceived threat to themselves and the town (self: OR = .22, CI$_{90\%}$ = .65–.73; town: OR = .17, CI$_{90\%}$ = .03–.65).

    Concerning the expected future damage caused by floods, results are inconsistent too. The RCS approach show a decrease

over time (RCS: OR = .65, CI$_{90\%}$ = .46–.91), while the panel does not show any change. In both rounds, respondents who experienced some sort of damage during the 2018 flood were more likely to report a higher expected future damage compared to those who did not experience any damage (Panel: OR = 10.30, CI$_{90\%}$ = 5.53–19.21, RCS: OR = 4.86, CI$_{90\%}$ = 2.94–8.11).

### 4.1.2 Knowledge and trust

Results about changes in knowledge are somewhat inconsistent across the two approaches. In round 2, the RCS shows that more male respondents feel like local knowledge (i.e. knowledge and information coming from relatives or friends) contributed to their knowledge of floods, compared to round 1 (OR = 1.86, CI$_{90\%}$ = 1.11–3.14), while the panel approach shows no difference over time. When it comes to knowledge deriving from official information, the RCS shows that it did not change over time, and this could be due to the fact that only a small fraction of respondents in the second round (9%)

took part in the informative events hosted by the municipality. A possible explanation of such a low turnout is that the respondents in the second round had not been interviewed the year before, and thus may have been less aware of the municipality's activities in relation to the flood event. On the other hand, the panel approach shows that respondents' knowledge of floods thanks to information received from official sources increased in the second round (OR = 2.78, CI$_{90\%}$ = 1.73–4.46). This may explain the lack of changes in perceived threat posed by floods in the panel dataset, and two factors

support this hypothesis. First, respondents who participated in the informative events organised by the local administration (24%) are more likely to report higher levels of knowledge because of having received official information (OR = 9.18, CI$_{90\%}$ = 3.93–22.44). Second, panel respondents in the second round tend to trust the local administration more. Panel respondents' trust in terms of the administration's risk communication is higher in the second round (OR = 2.34, CI$_{90\%}$ = 1.48–3.68). In particular, a) respondents who suffered low damage, and b) older respondents show an increase in trust in the

administration's risk communication (a: OR = 3.54, CI$_{90\%}$ = 1.22–10.23, b: OR = 1.03, CI$_{90\%}$ = 1.00–1.06). In the second round, panel respondents who participated in the informative events organised by the local administration show a higher trust





than those who did not (OR = 3.02, CI$_{90\%}$ = 1.37–6.83). No age differences were found in the panel dataset in terms of knowledge.

Contrary to knowledge, trust shows consistent trends over time across the two approaches. Panel respondents' trust in the

local administration concerning flood protection also increased (OR = 3.77, CI$_{90\%}$ = 2.27–6.26). It increased especially for those who suffered low damage during the 2018 flood (OR = 3.5.94, CI$_{90\%}$ = 2.73–12.94), and more in women, compared to men (OR = 3.12, CI$_{90\%}$ = 1.15–8.48). However, because in the first round they had a lower trust than men in the local administration concerning flood protection (OR = .58, CI$_{90\%}$ =.35–.96), we can say that it now is almost equally high for men and women. Similarly to the panel, The RCS approach also shows that women's and respondents who suffered low

damage's trust in the local administration when it comes to flood protection increased, compared to round 1 (women: OR = 1.81,  CI$_{90\%}$ = 1.12–2.92, low damage: OR = 2.00, CI$_{90\%}$ = 1.13–3.58). RCS respondents' trust in the local administration's risk communication does not seem to change over time, but if we break the sample down according to damage suffered during the 2018 flood, we notice that it actually increased, but only for those who suffered low damage (OR = 1.81, CI$_{90\%}$ = 1.04–3.15). RCS respondents who experienced high damage during the 2018 flood (4 or 5 on the 1 min 5 max scale) were

more likely to participate in the informative events (OR = 8.38, CI$_{90\%}$ = 2.50–32.28). They were also more likely to report a higher perceived threat to their house, in both years (OR = 23.02, CI$_{90\%}$ = 10.30–55.27), which may indicate why they participated in the informative events.

### 4.1.3 Public structural flood protection

To further explore changes in awareness from round 1 to round 2, we analysed respondents' knowledge and attitudes

towards the structural protection works undertaken by the municipality.

The panel results show that the majority of respondents (60%) knows about their existence, while this is true for only close to half on the respondents in the RCS (48%). In both, the majority found out about them on their own, e.g. driving by on their way to work (49% in the panel and 74% in the RCS). In the panel, an additional 29% found out through the local administration, while only 8% in the RCS. This difference may be due to the low participation rate to informative events of

respondents in the RCS. Despite this, both RCS and panel respondents in round 2 show a positive attitude towards public structural flood protection, as shown in Figure 2. This may additionally explain why respondents' risk awareness did not change significantly compared to the first round.

### 4.2 Preparedness

At a first glance, changes in perceived preparedness seem to be inconsistent across the two different longitudinal approaches.

In the RCS, no changes in preparedness were detected over time, while the panel results show a general increase in individual preparedness (OR= 3.48, CI$_{90\%}$ = 2.09–5.80). However, if we cluster the respondents depending on the amount of



damage suffered in the 2018 flood, we see that in the RCS preparedness actually increased only for those who experienced low damage (OR= 2.50, $CI_{90\%}$ = 1.40–4.54), and in the panel for those who experience no and low damage (no damage: OR = 2.64, $CI_{90\%}$ = 1.23–5.67; low damage: OR = 5.73, $CI_{90\%}$ = 2.42–13.61). This common result brings further evidence to the
fact that experiencing a flood with a low impact may promote a (sometimes false) sense of preparedness in the individual.

### 4.2.1 Private protection measures

The majority of respondents in both the RCS and the panel did not adopt any private structural protection measure (71% and 62% respectively), and of these, most do not intend to adopt any in the future either (65% and 70% respectively). In both cases, the main reason for not adopting any structural protection measure was the belief of living in a safe area. Respondents
who reported not having adopted any structural protection measure also reported lower levels of individual preparedness, compared to those who adopted such measures either before or after the 2018 event, both in the panel and in the RCS (Panel: OR = .32, $CI_{90\%}$ = .15–.64; RCS: OR = .25, $CI_{90\%}$ = .14–.43).

We then tested whether the panel respondents who experienced no or low damage were overrepresented in the group of respondents who adopted structural protection measures, as this could partly explain why they now feel more prepared (see
Figure 3). Given the categorical nature of the two variables, we ran Chi-squared tests to check for statistically significant differences between groups. Respondents who experienced high damage replied differently when asked about the adoption of structural protection measures compared to those who did not experience damage ($X^2$ = 20.95, p < .001) and to those who experienced low damage ($X^2$ = 9.60, p < .01).

### 4.2.2 Insurance

The relative majority of respondents both in the panel and in the RCS did not stipulate an insurance (respectively 40% and 51%). The reasons for not doing so include, in both cases, the belief of living in a safe area, the (perceived) rarity of flood events in the area, the high insurance costs, or legal issues (e.g. in case of renters, the insurance policy is the landlord's responsibility).

As opposed to RCS, in the panel approach having experienced damage in the 2018 flood affected positively the decision
about adopting an insurance policy. Here too, we tested whether the panel respondents who experienced no or low damage were overrepresented in the group of respondents who stipulated an insurance. Respondents who experienced high damage replied differently when asked about the stipulation of an insurance compared to those who did not experience damage ($X^2$ = 11.33, p < .01). The majority of respondents who experienced severe damage eventually stipulated an insurance after the event, while this number decreases as the damage suffered decreases (see Figure 4). However, some of the respondents who
suffered low damage already stipulated an insurance before the event, and even more so in the case of respondents who did not experience any damage.

As mentioned earlier, panel respondents who adopted private structural protection measures before or after the event report a higher individual preparedness than those who did not, but no such effect was found for those who stipulated an insurance.

### 4.3 Self-assessing changes in risk awareness and preparedness

In the second round, respondents were asked to self-assess how their risk awareness changed compared to the year before. In the panel dataset, half of respondents indicated an increase (49%) and the other half indicated no change (48%), while only 3% indicated a decrease. However, the self-assessment does not always match with the actual registered change in the answer given (see Figure 5). This often-sharp contrast may be due to the respondents not remembering their answer in the first round, and potentially interpreting the scale differently in the second round. However, with no evidence in this regard, it

is nearly impossible to determine the exact reason.

Concerning the self-assessed change in individual preparedness, in the panel 51% of respondents report an increase, 3% reports a decrease, and 46% reports no changes. This is the variable with the smaller gap between actual and self-assessed change, and it may be due to the ease with which one can assess their own preparedness, compared to a more abstract concept such as awareness.

In the RCS, the majority (66%) indicates an increase in their risk awareness, 33% indicates no change, and only 1% indicates a decrease. As for preparedness, the majority (59%) indicates no change, 37% think their preparedness increase, and 4% think it decreased. However, being the respondents in the two rounds different, it is not possible here to confront this result with any actual change in responses concerning risk awareness or preparedness.

### 5 Discussion

In the previous sections, we presented two methods to collect longitudinal data, i.e. with a RCS approach and with a panel approach, and the respective results. Here we argue that consistent results about the change (or lack thereof) in risk awareness and preparedness provide robust data to be employed in human-water systems modelling as well as in policy decision support.

### 5.1 Temporal dynamics

On average, risk awareness does not show significant changes over time, and this result is generally consistent across the two methods. However, when breaking down the sample to account for differences in terms of e.g. gender, or damage suffered, we see that certain variables evolve differently for different groups of individuals. For instance, women tend to have a higher perceived threat compared to men few months after the event, in round 1, but then their threat perception decreases over time, while men remain rather stable. Decreasing awareness in women is associated with their increasing trust in the local



administration and flood protection works. This was previously hypothesised by Viglione et al. (2014). In general, the lack
of changes in risk awareness can be explained by three aspects: a) the majority of respondents in the panel dataset
participated in informative events organised by the municipality; b) respondents in both the panel and the RCS dataset show
positive attitudes towards the public structural flood protection undertaken by the municipality; and c) relatively short time
elapsed between the two survey rounds (i.e. 12 months).

The first point, besides shedding light on why risk awareness did not change, brings evidence in favour of effective risk
communication strategies and community engagement, as was previously shown by Bodoque et al. (2019). They too found
that respondents who were exposed to risk communication activities maintained a rather stable level of risk awareness. An
effective risk communication strategy may provide a realistic view on the risk, where the awareness does not decrease
because the person is kept aware of the potential threat in terms of magnitude and likelihood, and it does not increase
because the person is provided with tools to deal with it in the future.

The second point touches upon the feeling of safety derived from the presence of public structural flood protection. This
theme has been widely discussed in the literature (Burby, 2006; De Marchi & Scolobig, 2012; Di Baldassarre, et al., 2018;
Ludy & Kondolf, 2012; Scolobig & De Marchi, 2009; Tobin, 1995; White, 1945), and is commonly referred to as safe-
development paradox (Kates et al., 2006). The presence of public structural flood protection may give the residents a false
sense of security, and often promotes urban development in areas at risk. In this instance, 73% of respondents in the panel
and 68% in the RCS dataset either agree or strongly agree with the statement "Public structural flood protection *eliminates*
the possibility of severe damage".  Such positive attitude towards the newly built public structural flood protection may
further explain why in the panel study awareness did not decrease, but perceived preparedness increased.

The third and last point provides additional insights into the time factor when it comes to risk awareness decay. Previous
studies adopting a longitudinal approach with longer time spans between survey rounds (e.g. Bubeck et al., 2020) could
capture more changes in awareness than our study. Moreover, other research results suggest that the decay of flood risk
awareness over time may range between a few years (Di Baldassarre et al., 2017) and a couple of generations (Fanta et al.,
2019). These results are, however, based on proxy data. Di Baldassarre et al. (2017) used flood insurance coverage in
California (Hanak et al., 2011), peaking after the 1997 Central Valley flooding, while Fanta et al. (2019) used archaeological
information about human settlements in Check Republic and changes in their vertical distance from the river before and after
major flood events.

Considering all these results, we can assert that an optimal time frame for conducting longitudinal studies to explore changes
in flood risk awareness would be up to one year after the flood event for the first survey round, and at least two years
between consecutive survey rounds. In absence of consequent flood events, this set-up allows for capturing changes over
time by avoiding to excessively zoom in or out. A similar time frame was also recently proposed by Seebauer and Babcicky



(2020), who argue for at least 1.5 years between survey rounds. However, these are just indicative time frames, and the specific context of the study area should always be considered. If a risk communication strategy is implemented, or another event occurs, the time frame should be adjusted accordingly.

Finally, our analysis shows a stable awareness and an increase in perceived preparedness (which can be interpreted as an
increase in coping appraisal) in respondents who experienced low damage during the 2018 flood, both in the panel and in the RCS. This result indirectly supports the Risk Perception Paradox, described by Wachinger et al. (2013). The paradox lays in the fact that people who experience a flood event with little consequences tend to have a lower risk awareness than those who did not experience an event or experienced it with severe consequences, and was previously reported in a number of other studies (Deeming, 2008; Green et al., 1991; Mileti & O'Brien, 1993; Wachinger & Renn, 2010). Here, while
awareness did not decrease, perceived preparedness increased, thus showing that the paradox concept is still valid.

**5.2 Methodological comparison**

Table 4 shows a summarised comparison of the two methods in terms of results. It is particularly relevant that both approaches show that, in the first survey round, women are more concerned than men when it comes to perceived threat to self and town. This is in line with previous studies on risk awareness and risk perception, where women were found in
general to be more concerned than men not only when it comes to floods (Cvetković et al., 2018), but also for other hazards such as e.g. road accidents (Cordellieri et al., 2016), or health risks (Galasso et al., 2020; Kim et al., 2018). In literature, this is known as the 'white male' effect (Finucane et al., 2010). In this sense, differences are not biological (or at least not entirely) but may lay on sociopolitical factors such as power and status. Results on the expected future damage are not consistent across the two methods, but the panel approach provides useful insights in the matter. While in the first round
there were statistically significant differences between those who experienced damage and those who did not in how they perceived potential future damage, these differences disappeared in the second round. This shows that risk awareness might change differently depending on damage suffered. When it comes to knowledge, results are not consistent across the two methods either. However, concerning trust, both approaches show that, in the second round, women seem to trust the local administration more, compared to the first round, and the same is valid for respondents who suffered low damages.

When it comes to individual preparedness, a first general glance at the two samples shows inconsistent results across the two methods. However, if we break down the respondents according to the severity of damage suffered, both approaches show that the perceived individual preparedness of respondents who experienced low damage in the 2018 increased in the second round.

These findings point us in the direction of improving the representation of socio-demographic heterogeneity in
sociohydrological flood risk models. The longitudinal data presented here shows that perceptions change differently over



time not only in men and women, but also depending on the severity of damage suffered in the past. Grouping individuals in sociohydrological models depending on certain characteristics, such as gender or previous experiences, constitutes a middle ground between a system dynamics and an agent-based modelling (ABM) approach. Indeed, it allows for embracing – at least partially – social diversity, while not completely losing the lumped approach which makes models generalizable and user-friendly. In fact, such compromise would make sociohydrological models appealing for policymakers because it would point out macro-scale differences within the community, thus highlighting potential weak links of existing risk communication strategies.

While in general a RCS approach provides data that can be employed in the classical system dynamics modelling using a lumped society, a panel approach yields valuable data to be employed in more case-specific modelling techniques, such as ABMs. However, if results from the two approaches are consistent, they can be employed for a more robust parameter estimation in both modelling techniques.

**6 Conclusions**

Over the past decades, the study of coupled human and water systems in various research fields and disciplines adopted a "system thinking" approach to embrace the complexities and underlying uncertainties of natural systems. Nonetheless, this is not always reflected in the way in which these systems are modelled. In system dynamics models of flood risk, for instance, society is often represented as a homogeneous group of individuals who act, react, and think in the same way. In absence of empirical longitudinal survey data, modelled dynamics of societal aspects such as risk awareness and preparedness may be flawed. As a result, models may lose their purpose of theorizing human-water interactions, as well as informing decision-makers in flood risk management.

This study provides insights not only in terms of attitude and behavioural change over time, but also on how these data should be employed in sociohydrological flood risk models. Results that are consistent *across* methods constitute robust data that can be employed for parameters' estimation, especially of those key variables with the highest uncertainty, such as the decay rate of flood risk awareness and preparedness. Our analysis shows a limited change in case of short analytical time frames and in the absence of events. Risk awareness remained stable for men but tended to decrease for women. Preparedness, on the other hand, only increased for those respondents who suffered low damage in the 2018 flood. The study also shows the need to enhance the representation of social diversity and processes in modelling human-water systems in general.

Limitations of this study are mainly related to short time frame of analysis, sample size and data collection. While surveying residents face-to-face is a way of establishing trust and it is essential (considering Data Privacy regulations) for targeting households on the basis of hazard assessment/risk exposure, it is highly resource- and time-consuming. One way to minimise



these issues is to conduct computer-assisted telephone interviews (CATI) or online surveys repeated over time (e.g. every 1.5 year). Among others, this can help reduce attrition/retention bias by starting off with much larger sample sizes and to collect reliable data about temporal dynamics of risk awareness and preparedness.

**Data availability**

The datasets supporting this research will be stored open access as .CSV on Zenodo upon acceptance of the manuscript, together with the survey forms (the original, in Italian, and the English translation) and a .xlsx file that provides additional information on all the variables in the datasets.

**Author contribution**

Conceptualisation: EM, AS, MB and GdB; Funding acquisition: GdB, EM, and MB; Data collection: EM; Formal analysis:
EM; Supervision: GdB and AS; Visualization: EM; Writing – original draft preparation: EM; Writing – review and editing: EM, AS, MB, and GdB.

**Competing interests**

The authors declare that they have no conflict of interest.

**Acknowledgements**

The Liljewalchs Travel Scholarship and the CNDS Interdisciplinary Grant partly funded the 2020 field trip. Part of the survey questions have been developed in the EC Sixth Framework Programme funded project FLOODsite, 2004–2008 (**http://www.floodsite.net**) Contract GOCE-CT-2004-505420. At the time of the FLOODsite research project Anna Scolobig was associated with ISIG (Institute of International Sociology of Gorizia, Italy), one of the FLOODsite partners. We wish to deeply thank the—at the time of FLOODsite research—leader of the Mass Emergency Programme at ISIG,
Bruna De Marchi for having considerably contributed to the survey design. The same is true for Professor Giovanni Delli Zotti and Maura Del Zotto, two of the other ISIG team members. We also thank all the other colleagues who provided us with professional advice and collaboration, and the interviewers who helped conduct the survey, Mattia Balestra, Giacomo Bernello, Giulia Bisoffi, Viviana Bort, Giovanna Caramuta, Fiorella Coco, Tania Di Mascia, Antonio Pica, Elena Poli, Luca Pressi, Federico Professione, and Niki Rigo. We also want to acknowledge the municipality of Negrar for their collaboration
and for providing demographic data, especially Lorenzo Calabria who considerably helped in bridging the interviewers with the residents.





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




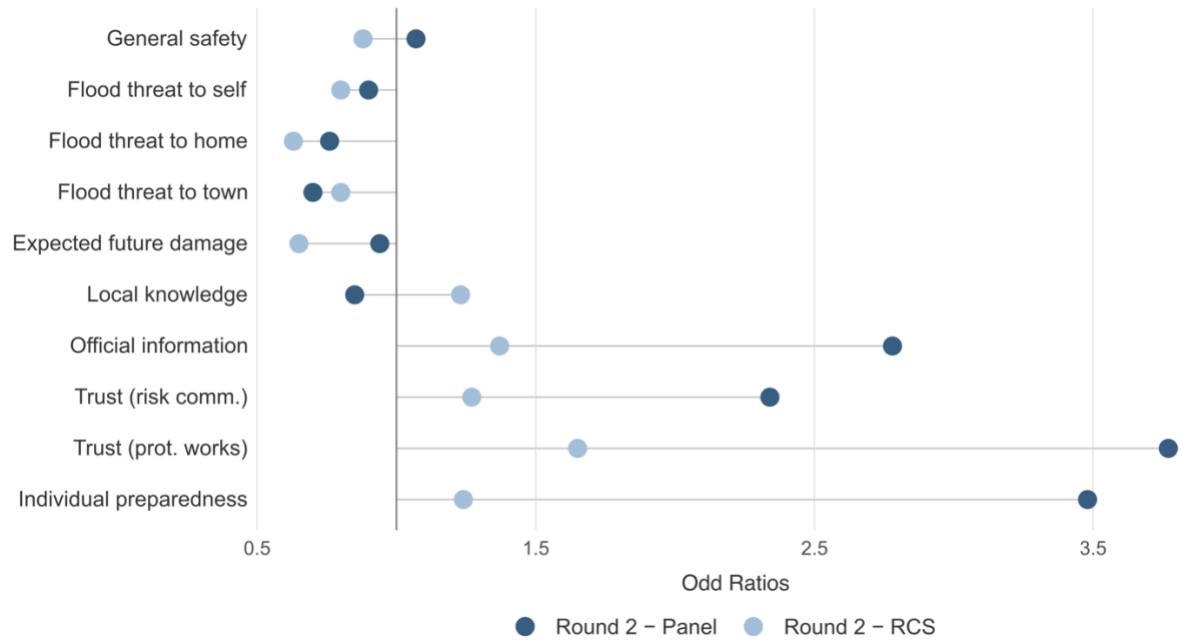

**Figure 1. Comparison of results from the two approaches on the effect of time on the variables of interest.**

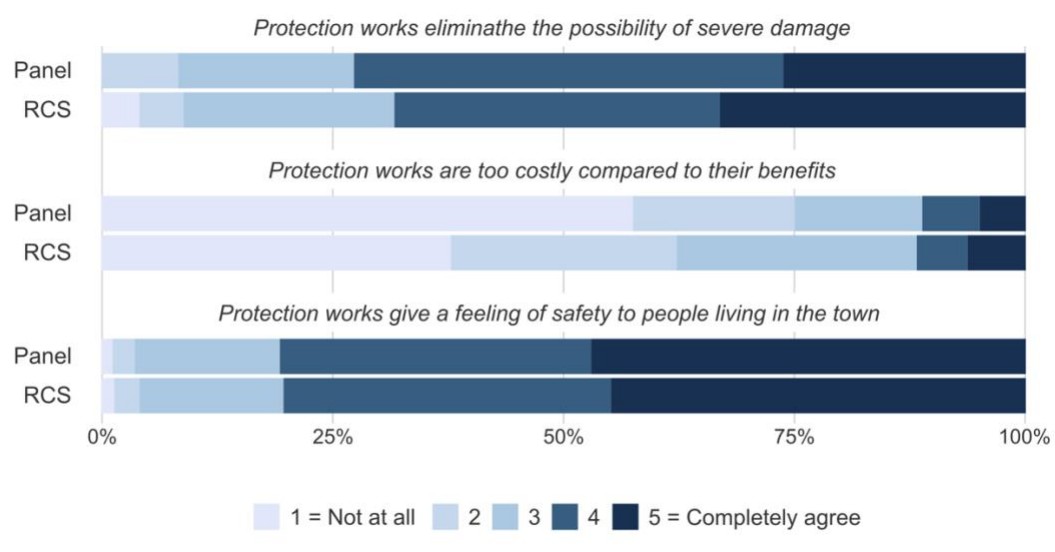


**Figure 2. Survey results from the second round on public structural flood protection undertaken by the municipality.**





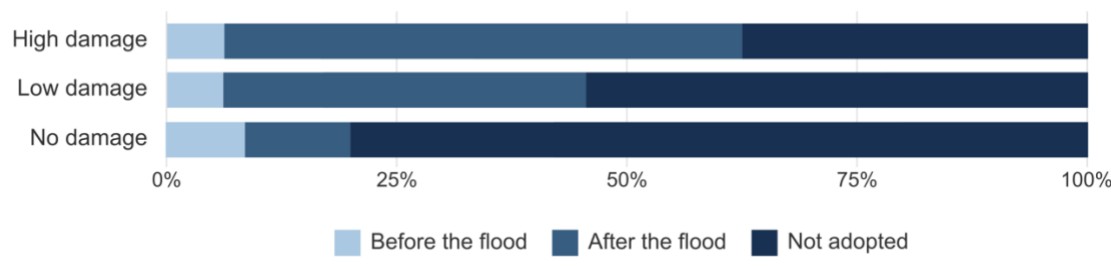

**Figure 3. Percentage of panel respondents who adopted private structural protection measures before the flood, after the flood, or did not adopt them, grouped by damage suffered in the 2018 flood.**


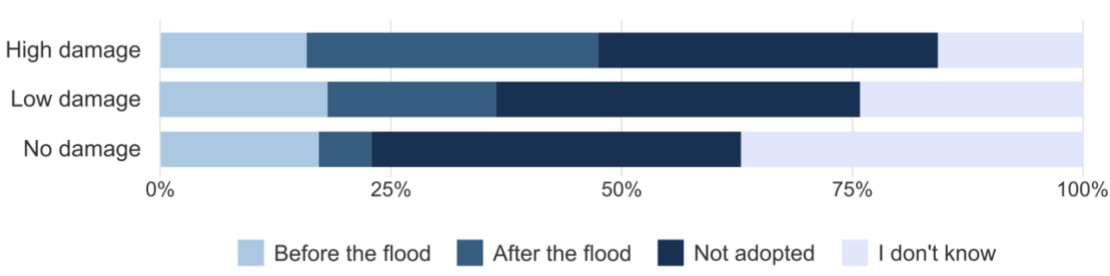

**Figure 4. Percentage of panel respondents who stipulated an insurance before the event, after the event, or did not stipulate any, grouped by damage suffered in the 2018 flood.**


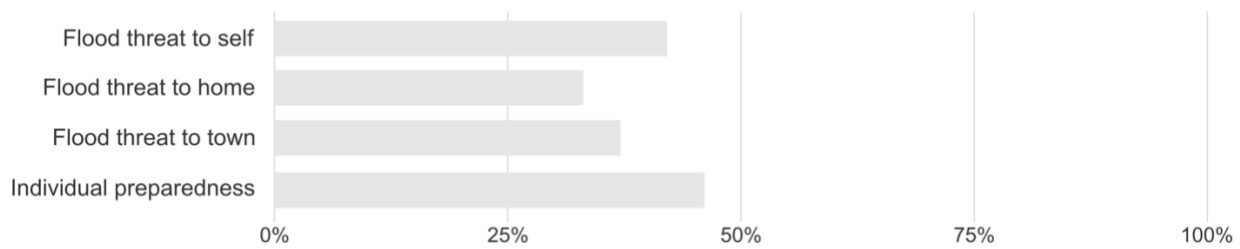

**Figure 5. Percentage of panel respondents for whom the perceived change matches the actual change.**






**Table 1. Summary statistics of the three samples collected.**

| Samples | N | Males | Females | Age (min) | Age (max) | Age (M) | Age (SD) | Experienced damage due to the 2018 flood | Participated in informative events* |
|---|---|---|---|---|---|---|---|---|---|
| Round 1 | 146 | 47.3% | 52.7% | 20 | 89 | 53.38 | 17.99 | 56% | – |
| Round 2 – RCS | 150 | 50.0% | 50.0% | 19 | 88 | 52.00 | 18.64 | 53% | 9% |
| Round 2 – Panel | 84 | 50.0% | 50.0% | 23 | 82 | 53.86 | 15.74 | 63% | 24% |

* This question was not asked in the first survey round

**Table 2. Main variables employed in the analysis.**

| Variable | Question | Available answers* |
|---|---|---|
| **General safety** | *To what extend does living here in this town make you feel safe?* | On a scale from 1, "Minimal safety" to 5, "Maximum safety", or "I don't know" |
| **Flood threat** | | |
| Perceived threat to self | *Considering floods, to what extent do you think they represent a threat to yourself personally?* | On a scale from 1, "Not at all a threat" to 5, "Serious threat", or "I don't know" |
| Perceived threat to home | *Considering floods, to what extent do you think they represent a threat to your home?* | On a scale from 1, "Not at all a threat" to 5, "Serious threat", or "I don't know" |
| Perceived threat to town as a whole | *Considering floods, to what extent do you think they represent a threat to the town as a whole?* | On a scale from 1, "Not at all a threat" to 5, "Serious threat", or "I don't know" |
| **Flood damage** | | |
| Damage experienced | *How severe was the damage you experienced during the 2018 flood?* | On a scale from 1, "No damage" to 5, "Serious damage", or "I don't know" |
| Expected future damage | *How much damage do you think a potential future flood could cause to your home?* | On a scale from 1, "No damage" to 5, "Serious damage", |





|  |  | or "I don't know" |
| --- | --- | --- |
| **Knowledge** | | |
| From local sources | *To what extent did knowledge from relatives and friends contribute to your knowledge about floods?* | On a scale from 1, "No contribution" to 5, "Great contribution", or "I don't know" |
| From official information | *To what extent did official information contribute to your knowledge about floods?* | On a scale from 1, "No contribution" to 5, "Great contribution", or "I don't know" |
| About structural flood protection | *Do you know of any structural flood protection in this area?* | 1, Yes<br>2, No<br>Or "I don't know" |
| **Trust in local administration** | | |
| On risk communication | *Should flood risk change in my area, the administration would inform me.* | On a scale from 1, "Completely disagree" to 5, "Completely agree", or "I don't know" |
| On structural flood protection | *I trust the local administration when it comes to structural flood protection.* | On a scale from 1, "Completely disagree" to 5, "Completely agree", or "I don't know" |
| **Preparedness** | | |
| Individual preparedness | *How prepared do you think you are to face a flood, in case it would occur?* | On a scale from 1, "Not at all prepared" to 5, "Highly prepared", or "I don't know" |
| Private protection measures | *Did your household adopt private structural protection measures?* | 1, Yes, before the 2018 event<br>2, Yes, after the 2018 event<br>3, No |
| Insurance | *Did your household adopt an insurance policy against floods?* | 1, Yes, before the 2018 event<br>2, Yes, after the 2018 event<br>3, No<br>Or "I don't know" |

\* "I don't know" answers were categorized as *NA* and excluded from the analysis




**Table 3. Logit regression model of the probability of a respondent moving from round 1 to round 2**

|  | Coefficients | Marginal effects |
|---|---|---|
| Age | 0,204*** | 0,049*** |
|  | (0,064) | (0,014) |
| Age squared | -0,002** | -0,000*** |
|  | (0,000) | (0,000) |
| Female | 0,245 | 0,059 |
|  | (0,410) | (0,101) |
| Suffered high damage | -0,123 | -0,030 |
|  | (0,176) | (0,042) |
| Experienced flooding before | 0,282 | 0,069 |
|  | (0,458) | (0,122) |
| Threat to self | -0,237 | -0,055 |
|  | (0,174) | (0,045) |
| Threat to house | 0,209 | 0,057 |
|  | (0,238) | (0,045) |
| Threat to town | -0,428** | -0,104* |
|  | (0,219) | (0,058) |
| Individual preparedness | 0,318* | 0,077* |
|  | (0,179) | (0,045) |
| Expected future damage | -0,042 | -0,010 |
|  | (0,260) | (0,065) |

N = 146
Robust standard errors in brackets
*** $p < 0.01$, ** $p < 0.05$, * $p < 0.1$


**Table 4. Summary of the main results and their robustness for application in human-water systems models. The column "Entire sample" shows the results from the linear regressions without interactions. The column "Significant interactions" shows the results from the linear regressions with interaction terms (gender or damage suffered).**

| Variable | Repeated Cross-Sectional | | Panel | | Robust |
|---|---|---|---|---|---|
| | *Entire sample* | *Significant interactions* | *Entire sample* | *Significant interactions* | |
| *Risk awareness* | | | | | |
| General feeling of safety | No change | Gender *(decreased in women)* | No change | – | No |
| Threat to self | No change | Damage *(increased only for respondents who suffered high damage, decreased for those who suffered no damage)* | No change | – | No |
| Threat to home | Decreased | Gender *(decreased for women)* | No change | – | No |
| Threat to town as a whole | No change | – | No change | Gender *(decreased for women)* | No |
| Expected future damage | Decreased | – | No change | – | No |
| Local knowledge | No change | Gender *(increased only for men)* | No change | – | No |
| Official information | No change | – | Increased | – | No |
| Trust in administration for risk communication | No change | Damage *(increased for those who suffered low damage)* | Increased | Damage *(increased for those who suffered low damage)* <br><br> Age *(increased for older respondents)* | Robust |
| Trust in administration for protection works | No change | Gender *(increased only for women)* <br><br> Damage *(increased for respondents who suffered low damage)* | Increased | Gender *(increased more for women)* <br><br> Damage *(increased for respondents who suffered low damage)* | Robust |
| | | | | | |
| *Preparedness* | | | | | |
| Individual preparedness | No change | Damage *(increased only for those who suffered low damage)* | Increased | Damage *(increased only for those who suffered no or low damage)* | Robust |
