# Peer review of "Longitudinal survey data for diversifying temporal dynamics in flood risk modelling"

_Natural Hazards and Earth System Sciences, 2021_

## Author Response (AR1)

In our response below, we will use *Rn.m to indicate the referee comment* and An.m to indicate the authors' reply, where n is the referee number and m the comment number. Line numbers in the authors' response refer to the manuscript file *without* track changes. In the manuscript file *with* track-changes, added text is in blue, deleted text is in , and moved text is in green.

**Referee 1**

*R1.1: Dear authors,*
*the paper with the title „Longitudinal survey data for diversifying temporal dynamics in flood risk modelling"*
*address a very interesting topic, not only in social science with the use of longitudinal methodological approaches to understand mid-term/long-term changes within a community in terms of flood risk management. The paper fits to the scope of the journal. I have some questions and remarks on the paper, which might need a larger more in-depth assessment of the current status of the paper. In overall, there are some major and minor remarks to the paper.*

A1.1 We again thank the Referee for taking the time to review our manuscript and for providing many constructive comments that helped us improve the description of our work.

*R1.2: A first overall remark reflects your conceptual framework, aim of the paper, discussion and literature review. The first point I would re-consider within the current version is the question about the use of theoretical framework. The paper needs to re-think to add theoretical-psychological framework (or frameworks) to assess and to explain your results, which theoretical concept (besides the socio-hydrology framework) are you using: Protective Action Decision Model (PADM), Risk Information Seeking and Processing model (RISP), Framework for Risk Information and Seeking (FRIS), Planned Risk Information Seeking Model (PRISM), Protection Motivation Theory (PMT), Transtheoretical model (TTM), Health Belief Model (HBM), Social amplification of risk framework (SARF), Model of Private Proactive Adaptation to Climate Change (MPPACC), Regulatory Focus Theory (RFT), Theory of Reasoned Action (TRA), or Community Engagement Theory (CET)? I would strongly suggest choosing one of these theoretical frameworks to re-consider your aim of the paper as well as your discussion.*

A1.2: We agree: the use of a theoretical framework is often useful to define the aim of a study. We clarified in the revised manuscript that our survey loosely refers to the Protection Motivation Theory (PMT) (lines 273-286). We also clarified that the main goal of our work is to conduct a methodological comparison between two longitudinal study approaches (lines 122-128). To avoid any confusion in the aim of our paper or interpretation of results, we removed the paragraphs about protective measures and flood insurance. These data, which were not collected longitudinally, will be employed in future studies exploring the relationship between risk awareness and preparedness.

*R1.3: What I'm largely missing is: what's the innovation in your paper, what's the theoretical added-value of your paper; I see large potential within the paper, but at the moment this isn't addressed, such as aim of the paper reads more like providing some new case studies; I largely missing a theoretical debate within your paper, which needs to be addressed within your aim of the paper.*

A1.3: We thank the Referee for highlighting that the aim of our paper was not stated clearly. We specified the aim clearly in the revised version of the manuscript (lines 122-128). Indeed, the innovation of this paper largely lays in the methodological contribution to the understanding of risk perception evolution/changes. As discussed above, our aim is to compare how results change when using two different longitudinal study designs in terms of sampling, specifically repeated cross-sectional and panel. In addition to this, we provide

longitudinal evidence in terms of flood risk awareness, trust, and perceived preparedness through an entirely new set of data.

*R1.4: This goes also hand in hand with your literature review, which included a wide range of papers from socio-hydrology community (which is fine), but none of these papers are really talking about risk perception (in terms of theoretical contribution!). Please, provide a broader risk perception debate within your paper.*

A1.4: We agree that we have provided limited background on flood risk perception, so we added a new section in the revised manuscript (lines 110-121) where we discuss previous literature on risk perception and its dynamics over time.

*R1.5: Another point, what are the differences of this paper with your already published work in Journal of Hydrology and Hydrological Sciences Journal?*

A1.5: This paper considerably differs from previously published work. This paper relies on an entirely new set of data based on two surveys carried out with alternative methods in a new case study (Negrar). Apart from being in the same country (Italy), this case study is different from the previous ones (Romagnano and Vermiglio) in terms of both type of hazard (debris flows vs. floods), temporality of the extreme event occurrences, as well as timing of the two surveys (1 year apart vs. 13 years apart). Moreover, the paper published in Journal of Hydrology adopts a probabilistic approach to the data analysis (using Latent Class Analysis), whereas the paper published in Hydrological Sciences Journal adopts a deterministic approach. While these first two papers aimed at exploring what influences the change of perceptions (or lack thereof), in this current paper, as discussed previously, we aim at comparing two different approaches for longitudinal studies (in terms of sampling methodology) to explore their effects on the dynamics of risk awareness.

*R1.6: Some minor points: also reconsider some structural aspects, such as within the discussion: you are talking about risk perception paradox, but this isn't mentioned in your introduction, similar aspect is the 'white male' effect; you should mention and explain this concepts already in your introduction.*

A1.6: We have now introduced these concepts earlier on in the introduction of the revised manuscript (lines 115-121).

*R1.7: Important issue: what's your theoretical added value to the current psychological debates on risk perception and coping strategy and adaptive capacity?*

A1.7: While our survey loosely refers to the Protection Motivation Theory, a theoretical contribution to the psychological debate is out of the focus of this paper. This comment makes us realize that mentioning adaptive behavior can create confusion. Thus, we removed the short results sections dedicated to this aspect that may distract from our main contribution: the methodological comparison between two different longitudinal approaches.

*R1.8: Second point goes directly to your data and used methodology: I would suggest providing much more detail information about the sampling: how you selected them, who drop out, why, how etc.*

A1.8: In the revised manuscript, we added more information on the respondents who dropped out (% females/males, age, damage suffered, education, and income, see line 307 and Table 3).

*R1.9: Also, I would re-think about your used methodology in analysing your data: you are using a very descriptive approach with some Chi-Square assessments: I would suggest providing a more sophisticated*

*statistical analysis, such as using a series of hierarchical regression models as there is much more potential within your dataset.*

A1.9: The Chi-square tests were used only for testing differences in answers to certain specific questions between respondents who suffered no-low-high damage, to additionally explain the results from the ordinal logistic regressions (which were at the center of our data analysis). Using hierarchical regression models would definitely be appropriate when testing a theory, but in this paper the data analysis focuses on exploring how certain variables change over time, thus there is only one predictor/independent variable (time). We then additionally tested for interactions of time with gender and amount of damage suffered, to explore whether it changed differently for different groups of people (lines 289-292).

*R1.10: In addition, the paper only address the social part of the socio-hydrology modelling framework. I would suggest including at least some information about the hydrology issue and risk analysis from the events, including social and physical vulnerability assessment of your sampling and selected case study. In particular, you mention a very interesting point, mainly low damage and preparedness. At the moment, it's largely unclear what does it mean low damage, low damage in terms of physical vulnerability, in terms of social vulnerability, in terms of a combination of both, see for example your statement on page 18, on the line 520? I would strongly suggest to include both vulnerability concepts in the revised version to understand your assumption about low damage/high damage; we need much more information about your sampling.*

A1.10: This comment highlights that this part was not sufficiently clear. Thus, the revised manuscript provides more information about the hydrology of the flood event and associated risks (lines 224-228). Second, regarding the damage, in Table 2 we report the specific questions upon which the variables are based. Damage suffered refers to damage suffered by the respondents and/or the respondent's home. This question was intentionally left vaguer to let the respondent interpret what they consider "damage", as it is their interpretation, not the researcher's, what eventually would influence their risk perception. Considering that there were no fatalities and no injured residents in the 2018 event, we consider this to be damage suffered mainly to the place of residence and belongings (e.g. car, motorcycle, etc.). Regarding the statement on page 18 line 520, in the revised manuscript we specified that we are actually referring to "perceived preparedness", to avoid any further confusion around the concept.

*R1.11: Third issue reflects your results section. First, the sections are too short; some of the sections only include 100-200 words, as NHESS has no upper word limited I would suggest to extend this part.*

A1.11: We agree and in light of the discussion above we removed these short sections referring to insurance and private protection measures in the revised manuscript, as they are not central to the aim of the paper.

*R1.12: Second issue reflects two key results sections: mainly, the private protection measures and insurance. First to the private protection measures: this is too far unclear what you are asking them: did you asked them to use PLFRA? Did you asked them to implement emergency management activities? Which measures did you asked them? This section needs more detail information about the type of measures, how they implemented it, the role of self-efficacy etc.?*

A1.12: We did not investigate which type of measures they implemented, neither how, nor their self-efficacy. It was a yes/no question on whether they adopted any private structural protection measure. Thus, we removed it from the paper.

*R1.13: Second aspect reflects your insurance section: as far I remember, the Italian government didn't introduced the Monti law about natural hazards insurance or is there any private or mandatory natural hazard*

*insurance system in Italy or is the government providing the disaster aid payments? If not, you probably will always get some wrong impression from your results on insurance like other surveys in other countries, such as Austria, Germany, Czech Republic etc. where people associate household insurance bill with a natural hazard insurance bill, which isn't correct as in Germany, for example, most insurance companies are not providing any compensation; the disaster payments are provided (usually) from the public administration. I think this might cause some misunderstandings for many readers across the globe (especially Anglo-Saxony countries) where the private insurance market plays a central role in the recovery. I'm not sure about this sections, it might be worth to remove it.*

A1.13: In light of potential confusion and due to the small amount of information on insurance collected through the questionnaire (here too, just a yes/no question), we removed this section from the paper.

*R1.14: Fourth issue goes to your statements about system dynamics and ABM. First of all, I don't think ABM might the solution to include individual risk perception or human behaviour aspect. As most (or almost most) ABM studies are using a stochastic approach to explain individual behaviour, main reason is that all humans are rarely acting within a swarm like in biology. Secondly, how you would integrate your results within current SD modelling frameworks, such as Vensim or Stella? It's more me a little bit vague how you argue; please provide a much more detail information about it.*

A1.14: We added more information on the way in which our results can inform the modeling of human behavior. Lines 534-544 discuss how different longitudinal data can be used in different models depending on the purpose of the model itself.

*Minor points:*
*R1.15: Please provide more in-depth information on table 1, such as population composition to understand how your sample reflects the population.*

A1.15: In the revised manuscript we included as much information as possible about the sample, including summary statistics on the education level and income of respondents (see lines 268-271 and Table 1).

*R1.16: Page 9, line 238 restriction to one person within the household: I see this as a very problematic issue as we know from previous studies the factor of gender in the rural-side play a crucial role in the data; I don't like this fact you exclude the others from the household. Therefore, I would suggest to add this aspect as a crucial limitations in your paper.*

A1.16: While we agree with the Referee that gender differences are an important factor in the risk perception literature, we disagree that interviewing one person per household represents a crucial limitation. The area is not rural, and in lines 253-256 we reported why we initially decided to interview one person per household. Many similar studies in the literature report interviewing only one person per household (see, among others, Bubeck et al., 2020; Kreibich et al., 2011; Osberghaus & Hinrichs, 2020; Seebauer & Babcicky, 2020), showing that it is a common practice. Moreover, we selected one person per household based on quotas including gender and age (in practice, the interviewers had a grid to fill in). As a result, our data is representative of the local population not only in terms of gender, but also of age. Unless each and every household member of each and every household in the study area were interviewed, the sample would have lost its representativeness. In addition, we extensively explore the role of gender and show the differences between men and women when it comes to risk perception.

Osberghaus, D., & Hinrichs, H. (2020). The Effectiveness of a Large-Scale Flood Risk Awareness Campaign: Evidence from Two Panel Data Sets. *Risk Analysis*. https://doi.org/10.1111/risa.13601

Bubeck, P., Berghäuser, L., Hudson, P., & Thieken, A. H. (2020). Using Panel Data to Understand the Dynamics of Human Behavior in Response to Flooding. *Risk Analysis*. https://doi.org/10.1111/risa.13548

Seebauer, S., & Babcicky, P. (2020). (Almost) all Quiet Over One and a Half Years: A Longitudinal Study on Causality Between Key Determinants of Private Flood Mitigation. *Risk Analysis*. https://doi.org/10.1111/risa.13598

Kreibich, H., Seifert, I., Thieken, A. H., Lindquist, E., Wagner, K., & Merz, B. (2011). Recent changes in flood preparedness of private households and businesses in Germany. *Regional Environmental Change*, *11*(1), 59–71. https://doi.org/10.1007/s10113-010-0119-3

*R1.17: Page 13, line 376: you are using cluster analysis: how did you organise the cluster analysis within this very small sampling? It's somehow unclear for me.*

A1.17: We clarified this aspect in the revised paper. What we meant with clustering was simply grouping the individuals depending on the amount of damage suffered: those who suffered no damage (replied 1 on the 1-5 scale), those who suffered low damage (replied 2-3 on the 1-5 scale) and those who suffered high damage (replied 4-5 on the 1-5 scale), to see whether risk perception dynamics were different depending on the amount of damage suffered (see lines 289-292).

*R1.18: Page 18, line 498: please provide some cross-references on the ABM section. Similar later on SD.*

A1.18: We added more references in the revised paper (lines 549-550).

Haer, T., Botzen, W. J. W., and Aerts, J. C. J. H., Advancing disaster policies by integrating dynamic adaptive behaviour in risk assessments using an agent-based modelling approach, *Environ. Res. Lett.*, 14(4), doi:10.1088/1748-9326/ab0770, 2019.

Liu, J. J. W., Reed, M., and Girard, T. A., Advancing resilience: An integrative, multi-system model of resilience, *Pers. Individ. Dif.*, 111, 111–118, doi:10.1016/j.paid.2017.02.007, 2017.

Michaelis, T., Brandimarte, L., and Mazzoleni, M., Capturing flood-risk dynamics with a coupled agent-based and hydraulic modelling framework, *Hydrol. Sci. J.*, 65(9), 1458–1473, doi:10.1080/02626667.2020.1750617, 2020.

Viglione, A., Di Baldassarre, G., Brandimarte, L., Kuil, L., Carr, G., Salinas, J. L., Scolobig, A., and Blöschl, G., Insights from socio-hydrology modelling on dealing with flood risk - Roles of collective memory, risk-taking attitude and trust, *J. Hydrol.*, 518(PA), 71–82, doi:10.1016/j.jhydrol.2014.01.018, 2014.

*R1.19: What I'm somehow missing are the role about the Italian civil protection in the case study; does the civil protection system in Italy influencing your preparedness as well as understanding of risk perception?*

A1.19: We added a section in the revised manuscript regarding the role of civil protection in the area (lines 222-223).

*R1.20: Finally, please provide some further limitations within your study as well as next research steps within the longitudinal research in flood risk management.*

A1.20: We added a section about future research within the flood risk domain (lines 575-577).

**Referee 2**

*R2.1: I find this paper is both an interesting and relevant contribution to the topics considered important to the audience of NHESS. This is especially regarding how we can improve socio-hydrological models, or other forms of climate and disaster risk adaptation.*

A2.1: We thank the Referee for their kind, positive, and constructive comments, which definitely contributed to improve our paper.

*However, I have several comments on the paper, that I hope to provide the constructive criticism required to push the paper forwards towards full publication in NHESS, especially as I am receptive to the central messages of the paper. Therefore, my main criticism of the paper is how the authors go about comparing the different data sets. I hope this sparks a fruitful discussion and seeing the revised version.*

    *R2.2.      I would perceive that the central value of the paper, and its useful extension of the scientific literature, is not on the results that they present per se, but rather the comparison that is conducted between the repeated cross-sectional analysis results and that for the longitudinal data analysis. Therefore, rather than presenting new 'results' the manuscript indirectly aims to be a methodological contribution to the literature. This is the added value that the authors wish to demonstrate. They then demonstrate this by comparing the differences in their results and determine 'good' results are those that are consistent across both sampling approaches. In my view, and to be a bit provocative, this requires additional justification or presentation. This is for several reasons. The first as the authors say, the purpose of RCS and Panels are different, panels follow people over time, while RCS look for changes in the population average. Therefore, in terms of how they can be integrated into socio-hydrological models (for example) would be quite different. Most of the argumentation presented in the paper tends towards questions and approaches that would require panel datasets to avoid the creation of the artificial relationships and dynamics in the data (e.g., Agent-based models). Therefore, while I would normally agree with the authors that consistent results across multiple sets of data/methods would be the best, I am not sure with the argumentation they have presented for why we need longitudinal data that a comparison of the results is the best way to 'sell' the data and the benefits of longitude data. As I would disagree that consistent findings can be used to parametrize socio-hydrological models but rather the result from the data that best matches the model to be developed. The consistent patterns in their case might be useful for models in their study area, but now I do not know how that would transfer to models outside of their study area for instance.*

    A2.2: We thank the Referee for bringing up a number of critical points. In the revised version of the paper, we expanded our discussion on the integration of different types of longitudinal data in different model types (lines 534-544). Indeed, which data can benefit which model depends not only on the structure of the model itself, but also on the model's purpose (e.g. descriptive of generic patterns vs. tailored on a specific case). We also expanded the discussion on the respective opportunities and limitations of the two approaches and resulting datasets (lines 499-533), and in turn on their respective potential for generalizability (lines 535-540). In the revised discussion we also reflected more on when and for which models converging results can be considered robust (lines 540-544).

    *R2.3.      Furthermore, on a similar note, I feel that an extended and more detailed discussion on the relative benefits and cons of each surveying method based on the authors' experiences would improve the discussion section. For instance, repeated cross-section is easier than panel datasets and this could provide a useful input for the dynamic modelling of population wide interventions and not polices aimed at specific people for example. A greater discussion of these relative merits could be an improvement. You have mentioned pros and cons within a specific example (notwithstanding the*

*previous comment of if it is a good idea to directly compare behavior in an RSC vs. Panel) but not on the overall pros and cons what are more widely discussed (from what I understand of the literature you present).*

A2.3: We agree with the Referee that pros and cons of the two approaches can be discussed more in depth. In the revised paper, we added a section in the discussion in which we presented and examined the benefits and drawbacks of the two approaches in general, not only in our specific context, as this will benefit readers more broadly (lines 499-533).

*R2.4.        Socio-hydrological models or socio-psychological models, in my experience, are quite dependent on local conditions. Therefore, the argumentations that risk perceptions overall did not hugely change could be the result of the short time span between the survey waves, or because of the interventions. The first rationale, I would argue, is more likely to be transferable (especially with in the context of papers the authors cite) than the second. So, the transferability of these results should also be mentioned because of how there were ongoing interventions in the study site.*

A2.4: We agree with the Referee. In the revised version, we discussed more openly about issues and potential for transferability of the two methodologies (lines 535-540).

**Referee 3**

*R3.1: The authors present a very nice discussion of the difference between longitudinal cross-sectional and panel data. I believe this is a valuable contribution to the existing literature.*

A3.1: We thank the Referee for their positive and constructive comments that contributed to improving the quality of our paper.

*R3.2: However, there is one major issue that I think should be addressed. In the introduction and discussion the authors frame the main contribution of this manuscript as a contribution to the sociohydrology literature and while I agree that there is a strong need for longitudinal data in sociohydrology, the main body of the manuscript does not really discuss the implications for sociohydrology. The main body provides a comparison of the two data sets, which in itself is very interesting, but is not really in line with the aim presented in the introduction and discussion. Therefore, I would suggest to either change the analysis and body of the manuscript to include the sociohydrology aspect, or to change the introduction and discussion to focus more on the difference between cross-sectional and panel data.*

A3.2: The Referee's comment points to a mismatch between the weight assigned to the concepts in the introduction and later on in the discussion of the original manuscript. In the revised version of the paper, we adjusted the weights of the different concepts (sociohydrology vs methodological contribution) in the introduction and discussion to make it more balanced.

*Some small points:*
*R3.3: In section 4.2 you need to better clarify when you are talking about perceived preparedness and when about the actual preparedness (i.e. the actual measures). For example the paragraph in lines 374-380 is a bit difficult to understand, it seems you are talking about perceived preparedness, but it would really help the reader, if you explicitly add perceived every time you are talking about perceived preparedness. Same in line 385, did respondents report lower levels of perceived preparedness or of actual measures implemented?*

A3.3: We thank the Referee for spotting a potential source of misunderstanding. In the revised manuscript, we made sure to use "perceived preparedness" whenever we refer to the respondent's self-assessment regarding their preparedness.

*R3.4: In section 4.2.1 you mention running Chi-squared tests and report that "Respondents who experienced high damage replied differently when asked about the adoption of structural protection measures". Different how? Please elaborate.*

A3.4: We added additional information in the revised manuscript to further explain the results from the Chi-squared tests (lines 404-410).

*R3.5: In section 4.2.2 you mention that "The relative majority of respondents both in the panel and in the RCS did not stipulate an insurance (respectively 40% and 51%)." But 40% is not a majority, or where there options other than having or not having insurance?*

A3.5: Indeed, there was the option of not knowing whether they were insured against flooding or not. We added this information and clarified this in the revised manuscript (lines 281-283).

*R3.6: In section 5.1 you mention the increase in perceived preparedness can be interpreted as a change in coping appraisal, but you have not explained the concept of coping appraisal, so I would either remove this comment or introduce coping appraisal earlier in the manuscript.*

A3.6: We agree with the Referee. In the revised manuscript, we rephrased this section so that no new concepts are introduced suddenly in the discussion and we introduced the concept of coping appraisal earlier on in the manuscript (line 275).

*R3.7: Some small language points:*
- *Abstract, line 35, times series should be time series.*
- *Line 84, I would suggest changing "community as homogenous community" into "a community as a homogenous community"*
- *Line 112, "the more robust is such data" should be "the more robust such data is"*
- *Line 140, acknowledge should be acknowledging*
- *Line 177, consider revising the sentence: "i.e. the number of observations is too small that any statistical analysis loses significance" You could replace too with so.*
- *Line 367, "half on" should be "half of"*
- *Line 399, "affected positively" should be "positively affected"*
- *Line 422, "being the respondents in the two rounds different" should be "the respondents in the two rounds being different"*
- *Line 460 do you mean the Czech Republic?*

A3.7: We thank the Referee for their careful proofreading of the paper and for spotting the mistakes above. We corrected all of them and proofread the entire paper.

---

## Author Response (AR2)

In our response below, we will use *Rn.m to indicate the referee comment* and An.m to indicate the authors' reply, where n is the referee number and m the comment number. Line numbers in the authors' response refer to the manuscript file *without* track changes. In the manuscript file *with* track-changes, added text is in blue, deleted text is in , and moved text is in green.

**Referee 1**

*R1.1: Dear authors,*
*thank you very much for your revised version. It reads very well, but I have a main comments: I'm still not 100% sure if your conceptual framework is the best suit once: I would rather recommend not to talk too much in detail about the use of socio-hydrology modelling as you aren't really providing information about how to improve the modelling work of socio-hydrology; I would rather suggest to focus your theoretical framework – which also has a stronger influence to your discussion/conclusion part – more on the current psychological debates on risk awareness / risk perception; meaning the different theoretical schools, such as PMT, SARF etc. and how your paper (in the discussion) are actually contributing to the current psychological schools.*

A1.1 We again thank the Referee for taking the time to review our manuscript and for providing additional comments that helped us improve our work. We have now worked on balancing our manuscript in the concluding section (lines 560–577), reserving more space for our theoretical contribution and removing the part concerning socio-hydrological modelling, as suggested. We also included limitations in the conclusions and added some directions for future studies.